# Is It Feasible to Preserve a Self-Sustaining Population of Yangtze Finless Porpoise in the Highest Density Section of Yangtze River?

**Weiping Wang** [1,2,†]**, Chongrui Wang** [3,4,†]**, Jinxiang Yu** [2,*] **and Bin Wu** [3,5,*]

1 Jiangxi Key Laboratory for Mass Spectrometry and Instrumentation, East China Institute of Technology, Nanchang 330013, China
2 Jiangxi Provincial Aquatic Biology Protection and Rescue Center, Nanchang 330000, China
3 College of Life Sciences, Nanjing Normal University, Nanjing 210023, China
4 Hunan Fisheries Science Institute, Changsha 410153, China
5 Fisheries Research Institute of Jiangxi Province, Nanchang 330000, China
* Correspondence: jxpync@126.com (J.Y.); wubinjx@163.com (B.W.)
† These authors contributed equally to this work.

**Abstract:** Using the VORTEX v. 10. 5.0.0, population viability analysis (PVA) was performed for Yangtze finless porpoise (YFP, *Neophocaena asiaeorientalis*) in the highest density section between Hukou and Meilong section (HMS) of the Yangtze River. Baseline model showed that this population was in a relatively vulnerability state; the deterministic growth rate (Det-r) was $-0.0230$; the stochastic growth rate (Stoch-r) was $-0.0385$; the probability of extinction (PE) was 0.5690; the mean population size of extant populations (N-extant) was 22; the genetic diversity (GD) was 0.7698. Under the general protection model, the Det-r was 0.0015, and the Stoch-r was $-0.0092$; Under the medium protection model, the Det-r was 0.0219, and the Stoch-r was 0.0144; Under the optimal protection model, the Det-r was 0.0383, and the Stoch-r was 0.0357. Sensitivity analysis found that adult females breeding rate, sex ratio at birth, and mortality rate of juvenile YFP were sensitive to maintaining population stability. The PVA showed that the conservation of YFP population in HMS depends on: neutralization of all threats affecting YFP population in the HMS; maintenance and, whenever possible, enhancement of the functional connectivity of the waterbody, increasing the food resources of YFP and reducing the risk of injury to YFP caused by human.

**Keywords:** population viability analysis (PVA); highest density; dispersal; gene flow; *Neophocaena asiaeorientalis*

## 1. Introduction

The Yangtze River is the mother river of Chinese. The Yangtze finless porpoise (YFP, *Neophocaena asiaeorientalis*) is a small, freshwater toothed cetacean, which is now considered as an independent species, population genomics of YFP reveal an incipient cetacean species adapted to freshwater [1,2]. It is an important indicator of the health of Yangtze River aquatic ecosystem. As the baiji (*Lipotes vexillifer*) is probably extinct [3,4], the YFP is the only cetacean to inhabit the Yangtze River. The populations of YFP in Yangtze River mainstream, Dongting Lake and Poyang Lake maybe have stable size and distribution. Only partial of them migrate from one region to another region along with the season change, which would drive the YFP population size of a region to fluctuate [5]. The newly revised List of Key State Protected Wild Animals in 2021 confirms that the YFP is officially promoted to the first class of protected wild animals in China [6]. The YFP has developed sonar system, which carries out life activities such as cruising, predation and communication through high frequency pulse signal and low frequency continuous signal. With the rapid social and economic development in the Yangtze River Basin, the quiet water environment is becoming less and less, the living space of the YFP is increasingly compressed. Increasingly

fragmented landscapes and the absence of sufficiently large high-quality protected areas are major problems for the YFP conservation [7–14]. The YFP is distributed in patches along the mainstream of the Yangtze River and has clear habitat selection [15]. However, periodic translocations of protected objects is an important intensive management approach to supplement the single viable population.

Population Viability Analysis (PVA) is intended to recommend conservation strategies [16]. Brook et al. [17,18] found that PVA predictions were surprisingly accurate. There are 5 PVA software packages, VORTEX [19], GAPPS [20], INMAT [21], RAMAS/Metapop [22] and RAMAS/Stage [23]. However, VORTEX has relatively few requirements for data, and the estimation results are relatively reliable. The model is widely used [24]. Recent studies have shown that the median time to extinction of YFP was 25–33 years in the Yangtze River [25]. Based on the analysis of population viability, researchers pointed out that under the comprehensive protection measures such as the prohibition of productive fishing, the population of YFP in the main stream of the Yangtze River will increase [26]. However, the YFP in the main stream of the Yangtze River belongs to fragmented distribution, so there is a large error in the analysis of the whole as a population, and there is only the significance of macro judgment of the population trend; The depth of understanding for each small population is inconsistent, and it is also difficult to accurately estimate the relevant analysis parameters. Therefore, selecting relatively independent and well-understood small populations for correlation analysis has strong guidance for the protection of regional small populations. At the same time, the model analysis results and the estimation of relevant parameters can also be verified and corrected in the continuous protection practice. Based on the high-precision, high-repeatability and high-coverage surveys of the representative YFP population, the dynamic trends of the population can be analyzed more effectively by using the representative model, formulate scientific protective measures [19].

## 2. Materials and Methods

A PVA was performed using field data [27] of population of YFP in the 180-km-long section of the Yangtze River, between Hukou and Meilong (HMS, as in Figure 1). The section has the highest porpoise density and abundance in all parts of the Yangtze River [28–30]. VORTEX 10.5.0.0 was used to model the viability of YFP population in HMS, mostly appropriate for analyzing extinction dynamics in small populations [24,31].

A compilation of all parameters used in the baseline model is presented in Table 1. The YFP population in HMS was modeled for 100 years. The generation length of YFP is estimated to be 8–9 years [26,32]; therefore, a 100-year simulation covers approximately 12 generations. To provide adequate precision, 10,000 replicates were run for each modeling scenario, the others were usually 1000 replicate simulations [33]. When minimum viable population was first discussed, a population having 99% probability to survive over 1000 years can be considered to have the minimum viable population size [34], therefore this analysis is also extended over 1000 y in the baseline scenario. Extinction was defined as the total removal of at least 1 sex [35]. The impact of inbreeding was modeled as 3.14 lethal equivalents (LE), with 50% of the effect of inbreeding ascribed to recessive lethal alleles [36]. The YFP's mating system is polygyny [37]. The mature age of female is 4–6 and that of male is 4.5–7 [37]. In 2015, a 134-centimetre-long female (4.5 years old) was sighted with a 107-centimetre-long female calf (0.4 years old) [38], according to the conversion of body length and age [39], the females of YFP may be able to reproduce at the age of 4. In this study, the average age of first-time offspring was 4 years for females and 5 years for males. According to the paternity test results of YFP in Poyang Lake, the females with body length of 152.5 cm in 2009 (ID no. 2009F7 in the paper) successfully gave birth to a male calf in 2015 (ID No. 2015M7 in the paper) [38]. According to the conversion of body length and age, the females of YFP may still be able to reproduce at the age of 18–19. In the present study, maximum age of reproduction of 18 years was set for both male and female YFP. No female YFP gives birth to more than 1 calf per litter/brood [40]. For this modeling, a maximum litter/brood size of 1 offspring was considered. The sex ratio at birth of YFP was

1:1 in Yangtze River Tian-e-zhou ancient channel [40]. YFP's sex ratio from 0 to 1 years of age was 1.1 in Poyang Lake (female: male ratio was 11:10) [38]. The sample size was limited and there was a possibility that this sample may not be fully representative of Poyang Lake population. For this model, the sex ratio at birth was 1:1. The inter-birth interval for YFP is close to 2 years [39]. In addition, some females may lose their offspring during lactation, or due to stillbirths or neonatal deaths and become available for reproduction sooner, reducing the inter-birth interval. For the baseline model, it was assumed that 50% of the sexually mature females would be reproducing in a given year. YFP is expected to show little variation, therefore, 10% EV, was used in the simulation. Density dependent reproduction was not included in the baseline model. It was considered that an average 70% of the adult males are capable of reproducing each year.

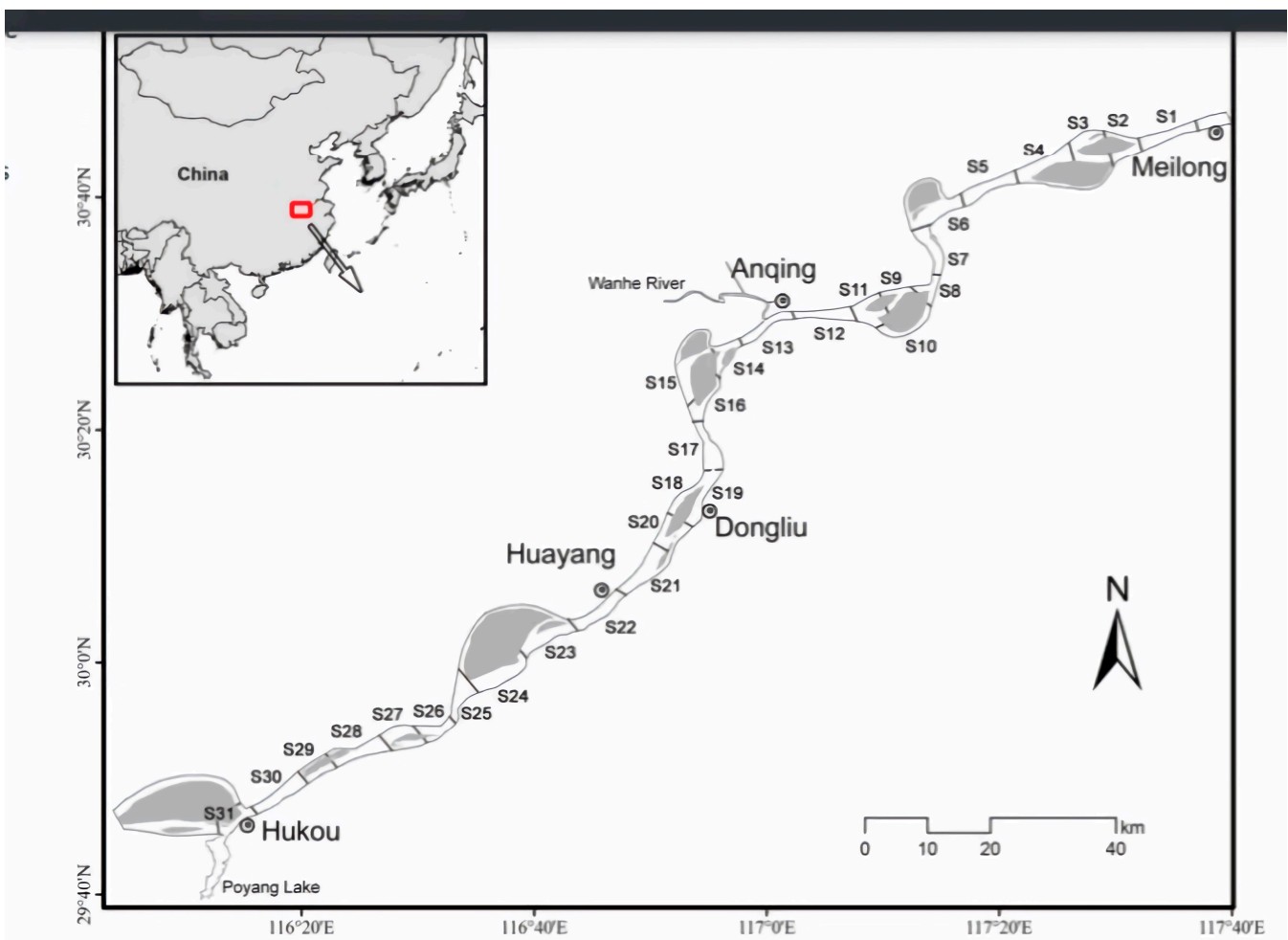

**Figure 1.** Map showing the distribution area of YFP in the 180-km-long section of the Yangtze River, between Hukou and Meilong.

**Table 1.** Summary of parameter input values used in the VORTEX baseline model for YFP population in HMS.

| Parameter | Baseline Value |
| --- | --- |
| Number of populations | 1 |
| Initial population size (*N*) | 181 |
| Carrying capacity (*K*) | 600 |
| Inbreeding depression | 3.14 LE |
| % of the effect of inbreeding due to recessive lethal alleles | 50 |
| Breeding system | Polygynous |

**Table 1.** *Cont.*

| Parameter | Baseline Value |
| --- | --- |
| Age of first reproduction by males/females | 5/4 years |
| Maximum reproductive age | 18 years |
| Annual % of adult females breeding (SD) | 50% (10%) |
| Density dependent reproduction? | No |
| Maximum litter size | 1 |
| Overall offspring sex ratio | 50:50 |
| Adult males in breeding pool (%) | 70 |
| Mortality rates: | |
| % mortality from age 0–1 (SD) | 25 (5) |
| % mortality from age 1–2 (SD) | 20 (5) |
| % mortality from other age (SD) | 10 (3) |
| Catastrophe | 1.75%; 50%, 50% 16.7%; 95%, 95% |
| Harvest | None |
| Supplementation | None |

According to Zhang et al. (1999) [32], the YFP mortality rate in the Yangtze River might be 20% for the 0–1 year age group, 20% for the 1–2 year age group and 15% for the other age groups. According to Li (2017) [40], the YFP 0–1, 1–2, 2–3, 3–4, 4–5, and >5 age groups mortality was 13.66, 12.37, 11.47, 10.92, 10.68, 13.15, in Tian-e-zhou semi-natural ex situ reserve. According to Yang et al. (1998) [41], the mortality rate of the YFP at the age of 0–2 years was 30.8%. According to Mei et al. (2012) [42], the mortality rate of the YFP at the age of 0–1 years was 22.4% −27.7%. For this model, the YFP 0–1, 1–2 and other age groups mortality was 25%, 20% and 10%. Standard deviations were set to 5% of the mean values for 0–2 age groups, and 3% of the mean values for other age groups, which seemed to provide a sensible interval. In the baseline model, only 1 population was considered, with no immigration or emigration of individuals. Population size in the HMS was 181 (95% CL125–239) animals on average [27]. According to Zhang et al. (1999) [32], the environmental capacity was 5000 of YFP in the Yangtze River. The HMS occupies <10% of the YFP distribution range, but nearly 40% of the YFP individuals in the Yangtze River inhabit this section [28]. The prediction results of the PVA model were credible when the maximum amount ever distributed in history was taken as the environmental capacity [18]. Preliminary estimation of environmental capacity of YFP in Poyang Lake through Yangtze River Waterway based on Ecopath model of the total ban on production fishing, showed that environmental capacity of YFP was 0.2 t/km$^2$, or about 625 heads of the waters [43]. To be conservative, thus, the environmental capacity was set 600 in HMS in the baseline scenario. The catastrophe per year probability was calculated simply from the catastrophe per generation probability using the following formula: $P(Y) = P(G)/T$ [44], where $P(Y)$ is the probability of a catastrophe per year, $P(G)$ is the probability of a catastrophe per generation (14%) and T is the generation length of the species (8 years for YFP). $P(Y) = 0.0175$. For this modeling, catastrophes were included with a per year probability of 1.75% causing a 50% decrease in survival and reproduction. The model excluded harvest and supplementation.

The sensitivity test module in VORTEX [19], was used in the sensitivity analysis to evaluate the effect of adult females breeding (AFB, 30%; 40%; 50%; 60%; 70%), mortality from age 0 to 1 (MA0, 10%; 12.5%; 15%; 17.5%; 20%; 22.5%; 25%; 27.5%) and age 1 to 2 (MA1, 10%; 12.5%; 15%; 17.5%; 20%; 22.5%), sex ratio at birth (SRB, 40%; 45%; 50%; 55%; 60%), initial population size (IPS, 125; 181; 239), carrying capacity (K, 300;600;1200), males in breeding pool (MBP, 40%; 70%; 100%), and catastrophes severity (CS, 0.3; 0.4; 0.5).

The baseline model was used to analyze different scenarios and conservation challenges facing YFP population in HMS. Annually capture and supplement a certain amount of 2–3 aged individuals with a sex ratio of 1:1 was set to simulate individual exchange. After the 2nd age of the Yangtze finless porpoise, the relationship between mother-calf pairs

is not very close. The exchange of the 2nd–3rd age porpoise can ensure the survival rate of the exchange individual, it can also minimize the effect of individual exchange on the reproductive capacity of the original population. Different numbers of immigration (2; 4; 6; 8; 10 and 12), emigration (2; 4; 6; 8; 10 and 12), individual exchange (8; 16 and 32) was used to represent different gene flow. Meanwhile, in order to reduce the direct impact of capture and replenishment, such migration and individual exchange should only take place when the population size was larger than 90. Three levels protection measures were tested, and their influence on the viability of the YFP population in HMS was analyzed. Under general protection, the survival of the population improved slightly (individual exchange, adult females breeding rate, and the mortality of 0–1 age, 1–2 age were set separately as 8 individuals of 2–3 age, 55%, 20% and 15%); under medium protection, the survival of the population improved even more (individual exchange, adult females breeding rate, and the mortality of 0–1 age, 1–2 age were set separately as 16 individuals of 2–3 age, 60%, 15% and 12.5%); under optimal protection, the survival of the population improved obviously (individual exchange, adult females breeding rate, and the mortality of 0–1 age, 1–2 age were set separately as 32 individuals of 2–3 age, 65%, 12.5% and 10%).

## 3. Results

### 3.1. Baseline Model

The baseline model resulted in a Det-r of –0.0230 ($\lambda$ = 0.9773). Generation time resulted in approximately 9 years for both sexes. The ratio of adult males to adult females was 0.878. The Stoch-r of this population in complete absence of catastrophe and threat was −0.0113, PE was 0.0307 and the mean population size of extant populations (N-extant) was 91, with 89.68% of the genetic diversity (GD) remaining in 100 years. When including catastrophe and threat, Stoch-r of this population was −0.0385, PE was 0.5690 and N-extant was 22, with 76.98% of GD remaining in 100 years. However, Stoch-r of this population was −0.041, PE was 1.0 and N-extant was 0, with 0 of GD remaining in 1000 years, The median extinction time was 95 years, and the extinction time 96.8 years. The rapid decline of the Yangtze finless porpoise population between 2006 and 2012 has been largely curbed, but the species is still critically endangered and conservation efforts need to be strengthened. As a result, millennium-scale assessments may currently have limited practical implications for the conservation of the YFP, therefore, subsequent analyses of this study were limited to 100 years.

### 3.2. Sensitivity Analysis

The percentage of adult females breeding rate, sex ratio at birth and mortality from age 0 to 1 yielded the highest sensitivity of Stoch-r, whereas for final population size (N-all), the results were similar. However, it is more sensitive to carrying capacity than mortality from age 0 to 1 (Figures 2 and 3).

### 3.3. Scenarios

Different amount of YFP migration to HMS had obvious influence on the Stoch-r, when the immigration reached 10, Stoch-r was greater than 0, the population size showed an increasing trend; when the emigration reached 8, Stoch-r was about −5%, the population size showed a rapid decline trend (Table 2). When the number of individuals exchanged goes from 0 to 8, Stoch-r gone ranges −0.0385 to −0.0367; when the number of individuals exchanged goes from 8 to 16, Stoch-r ranged −0.0367 to −0.0244; When the number of individuals exchanged goes from 0 to 16, Stoch-r gone ranges −0.0385 to −0.0244; When the number of individuals exchanged goes from 16 to 32, Stoch-r ranged −0.0244 to 0.0006, results were presented in Table 2. These results suggest the importance of maintaining a certain scale of gene flow, and the effect of gene flow level on population size was not a simple linear relationship, but an accelerated effect. Under the general protection model, the deterministic growth rate was 0.0015; the Stoch-r was −0.0092; PE was 0.1058 and N-extant was 186, with 92.65% of GD remaining; Under the medium protection model, the deterministic growth rate was 0.0219; the Stoch-r was 0.0144; PE was 0.0178 and N-

extant was 421, with 97.45% of GD remaining; Under the optimal protection model, the deterministic growth rate was 0.0383; the Stoch-r was 0.0357; PE was 0.0030 and N-extant was 522, with 99.26% of GD remaining (Table 2).

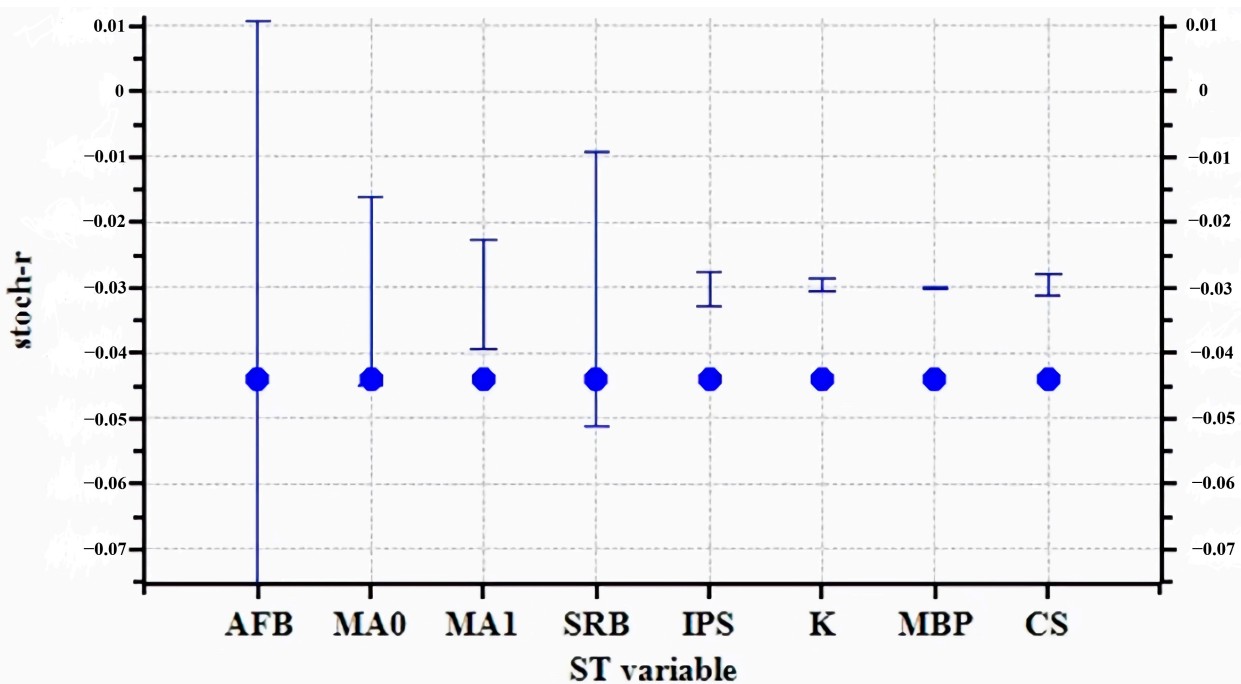

**Figure 2.** Sensitive analysis to stoch-r (AFB: adult females breeding; MA0: mortality from age 0 to 1; MA1: mortality from age 1 to 2; SRB: sex ratio at birth; IPS: initial population size; K: carrying capacity; MBP: males in breeding pool; CS: catastrophes severity. A Max-Min plot shown the range of results for each tested variable, averaged across the values of all other tested variables. A point was placed at the result for the base scenario. In a Max-Min plot, the variables with the longest lines had the greatest effect on results.).

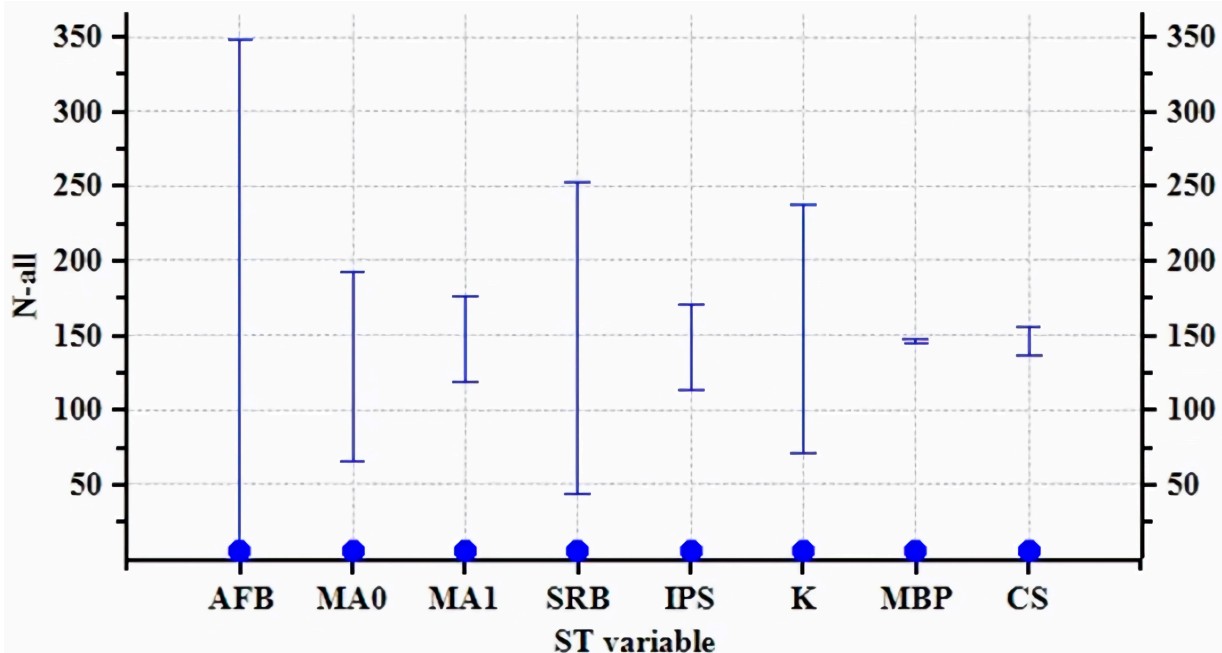

**Figure 3.** Sensitive analysis to N-all (AFB: adult females breeding; MA0: mortality from age 0 to 1; MA1: mortality from age 1 to 2; SRB: sex ratio at birth; IPS: initial population size; K: carrying capacity;

MBP: males in breeding pool; CS: catastrophes severity. A Max-Min plot shown the range of results for each tested variable, averaged across the values of all other tested variables. A point was placed at the result for the base scenario. In a Max-Min plot, the variables with the longest lines had the greatest effect on results).

**Table 2.** The deterministic growth rate (Det-r), stochastic growth rate (Stoch-r), probability of extinction (PE), the genetic diversity (GD) and the population size (N-extant) in 100 years under different scenarios.

| Projects | Det-r | Stoch-r | PE | GD | N-Extant |
|---|---|---|---|---|---|
| Model 1 (Baseline model) | −0.0230 | −0.0385 | 0.5690 | 0.7698 | 22 |
| Model 2 (Model 1 + immigration 2) | −0.0230 | −0.0273 | 0.3730 | 0.8694 | 68 |
| Model 3 (Model 1 + immigration 4) | −0.0230 | −0.0162 | 0.2440 | 0.9294 | 150 |
| Model 4 (Model 1 + immigration 6) | −0.0230 | −0.0068 | 0.1471 | 0.9603 | 241 |
| Model 5 (Model 1 + immigration 8) | −0.0230 | −0.0004 | 0.0972 | 0.9770 | 322 |
| Model 6 (Model 1 + immigration 10) | −0.0230 | 0.0043 | 0.0685 | 0.9848 | 384 |
| Model 7 (Model 1 + immigration 12) | −0.0230 | 0.0080 | 0.0519 | 0.9894 | 430 |
| Model 8 (Model 1 + emigration 2) | −0.0230 | −0.0438 | 0.6885 | 0.7171 | 15 |
| Model 9 (Model 1 + emigration 4) | −0.0230 | −0.0470 | 0.7555 | 0.6884 | 12 |
| Model 10 (Model 1 + emigration 6) | −0.0230 | −0.0497 | 0.7981 | 0.6766 | 12 |
| Model 11 (Model 1 + emigration 8) | −0.0230 | −0.0508 | 0.8209 | 0.6611 | 12 |
| Model 12 (Model 1 + emigration 10) | −0.0230 | −0.0517 | 0.8234 | 0.6716 | 12 |
| Model 13 (Model 1 + emigration 12) | −0.0230 | −0.0527 | 0.8422 | 0.6580 | 12 |
| Model 14 (Model 1 + individual exchange 8) | −0.0230 | −0.0367 | 0.5320 | 0.8050 | 29 |
| Model 15 (Model 14 + mortality rate of 0–1 age 20%, 1–2 age group 15%) | −0.0092 | −0.0212 | 0.2396 | 0.8838 | 86 |
| Model 16 (Model 15 + breeding rate 55%; general protection model) | 0.0015 | −0.0092 | 0.1058 | 0.9265 | 186 |
| Model 17 (Model 1 + individual exchange 16) | −0.0230 | −0.0244 | 0.3411 | 0.8977 | 88 |
| Model 18 (Model 17 + mortality rate of 0–1 age 15%, 1–2 age group 12.5%) | 0.0009 | −0.0057 | 0.0937 | 0.9438 | 232 |
| Model 19 (Model 18 + breeding rate 60%; medium protection model) | 0.0219 | 0.0144 | 0.0178 | 0.9745 | 421 |
| Model 20 (Model 1 + individual exchange 32 head) | −0.0230 | 0.0006 | 0.0706 | 0.9867 | 300 |
| Model 21 (Model 20 + mortality rate of 0–1 age 12.5%, 1–2 age group 10%) | 0.0074 | 0.0116 | 0.0218 | 0.9874 | 439 |
| Model 22 (Model 21 + breeding rate 65%; optimal protection model) | 0.0383 | 0.0357 | 0.0030 | 0.9926 | 522 |

## 4. Discussion

The potential annual growth rate of −2.3% provided evidence that even in the absence of additional catastrophes and threats, the YFP population in HMS will still decline. The result explains the low-density areas of YFP population from Yichang to Jingzhou, Xintan to Tuanfeng and Jiangyin to Shanghai throughout they have faced greater threats in the past, such as genetic isolation, pollution, illegal fishing, vessel traffic and construction of wading projects [14,45]. As a small regional population, the YFP population of 181 is likely to be very vulnerable in HMS over the next 100 years, the probability of extinction was 0.5690 and only 76.98% of genetic diversity remained. Studies of Poyang Lake's Yangtze finless porpoise also shown an effective population size of more than 200 individuals or a census population size of more than 1000 individuals is necessary, to realize the long-term goal of preserving more than 90% genetic diversity in 100 years [46]. Furthermore, the below zero stochastic growth rate means that the population was vulnerable to unforeseen threats, such as disease epidemic or an increase in a current threat, such as genetic isolation. However, a small population monitoring study also showed that, the population size of the YFP and the proportion of mother and child dolphins in the population showed a steady and increasing trend, indicating that the population of the YFP in Poyang Lake might have achieved positive growth [47]. Adult female breeding rate, sex ratio at birth and mortality from 0 to 1 years of age had the strongest influence on the dynamics of YFP population in HMS. The variability in ovulation rate as determined by corpora count may suggest that some females ovulate more often than others [48]. Therefore, it is an important task

to excavate and protect the fine breeding porpoise germplasm. Reproduction rate is the determining factor of how many descendants will exist in a population. In addition, if there is enough food, the breeding interval will be shortened [49]. Long-term studies have greatly enhanced the possibility for collecting relevant demographic information on long-lived, slowly reproducing cetacean populations in the wild [50].This means accurate monitoring of the number and proportion of mother-calf pairs in small populations of Yangtze finless porpoises is the basis for assessing the population dynamics, providing a good nursing environment for mother-calf pairs and reducing the risk of accidental injury is an important part of daily management and conservation. Habitat degradation and fragmentation pose a significant threat to the survival of the Yangtze finless porpoise. Future conservation research and practice should focus on habitat restoration of the solidified riverbanks to reestablish and enhance habitat connectivity [51]. Ecological corridors have the function of maintaining or restoring Ecological connectivity, and are important for connecting the biological habitats and protecting species diversity. The assessment results indicate that tourism and shipping activities are likely to be the main sources of impacts on the Indo-Pacific humpback dolphin ecological corridor, with the core corridor being more influenced by tourism activities and the secondary corridor being more influenced by shipping activities [52].

The modeling of the emigration reflected a reduction in gene flow for YFP population in HMS. These reduction in gene flow resulted in a decrease in the stochastic growth rate (See Model 8–13 in Table 2 for details). However, the modeling of the immigration and individual exchange reflected an increase in gene flow for YFP population in HMS. The regional connectivity, dispersal, and gene flow of small population with fragmented distribution might be determinant factor for its persistence [53–56] Translocations offer a good opportunity for management and should be explored as a useful tool for short-time rescue of small subpopulations at imminent extinction risk, but must be applied with caution [57]. Franklin (1980) [58] proposed the 50/500 rule. It still has a useful tool in conservation biology [59]. The population size of YFP population in HMS has been well below 500 individuals [27], indicating that there is severe genetic vulnerability of YFP population in HMS. In this study, we investigated potential conservation scenarios in three level protection cases where gene flow, adult females breeding rate and mortality rate of young YFP were all improved at several levels. The following protective measures can be taken, such as the improving habitat quality in HMS by enhancing the connectivity of YFP populations among different water bodies, increasing the food resources of YFP and reducing the risk of injury to YFP caused by human. With the rapid social and economic development in the Yangtze River Basin, the living space of the YFP is increasingly compressed. The underwater noise generated by frequent human activities will adversely affect its vocalization and hearing, and even cause physical injury [60]. As evidenced by the VORTEX model, the YFP population in HMS is vulnerable. Any threats affecting YFP must be neutralized. In the conservation scenarios, the declining trend of YFP population in HMS was changed to a stable and increasing trend. It is suggested to improve the living environment "small but comprehensive" by analyzing the sensitivity parameters, and formulate a complete set of protection schemes may be the main inspiration to us from the population viability analysis of YFP population in HMS. Hector's and Maui dolphins are predicted to recover without fisheries mortality [61,62] Fishing gear entanglement was also one of the main causes of accidental death of YFP [45], Thus, with the implementation of policies such as a total ban on productive fishing in the main stream of the Yangtze River, they could enhance the protection of YFP population [26]. Meanwhile, the population distribution and size of the porpoise based on passive acoustic monitoring can be employed as an important ecological assessment indicator to evaluate the eco-environmental quality and aquatic biodiversity, as well as their temporal-spatial changes along the river and in the lakes. the indicator could be used to evaluate the implementation of relevant protection work and the actual protection effect [63].

**Author Contributions:** Conceptualization, W.W. and B.W.; methodology, W.W. and B.W.; software, C.W. and B.W.; validation, J.Y. and B.W.; formal analysis, C.W. and B.W.; investigation, W.W. and B.W.; resources, C.W. and B.W.; data curation, W.W. and B.W.; writing—original draft preparation, W.W., C.W. and B.W.; writing—review and editing, J.Y. and B.W.; visualization, C.W. and B.W.; supervision, J.Y. and B.W.; project administration, J.Y. and B.W.; funding acquisition, the Jiangxi Provincial Agriculture, Animal Husbandry and Fisheries Guidance Project 2022. All authors have read and agreed to the published version of the manuscript.

**Funding:** This research was funded by the Jiangxi Provincial Agriculture, Animal Husbandry and Fisheries Guidance Project 2022, China, grant number [JXNMYY202242]. And the APC was funded by Jiangxi Key Laboratory for Mass Spectrometry and Instrumentation, East China Institute of Technology, Nanchang.

**Data Availability Statement:** The data that support the findings of this study are available from the corresponding author upon reasonable request.

**Acknowledgments:** The authors want to thank the Jiangxi Provincial Agriculture, Animal Husbandry and Fisheries Guidance Project 2022, China, for its support for this study.

**Conflicts of Interest:** The authors declare no conflict of interest.

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
