# Peer review of "Is It Feasible to Preserve a Self-Sustaining Population of Yangtze Finless Porpoise in the Highest Density Section of Yangtze River?"

_water, doi:10.3390/w15061215_

Round 1

Reviewer 1 Report

This MS can be improved following the points reported in the attached PDF file, in particular:

Try to make simulation over 1000y and, if possible, to determine the population size allowing a 99% probability of survive (Shaffer 1981) and include in the manuscript a curve of survival probability over time versus population size and report your comments on this figure in the manuscript.

Explain the acronyms used also in the legend of fig.2 (sensitivity analysis).

Check English and correct minor points reported in the revised PDF file.

Author Response

Thank you for your letter and for the reviewers’ comments concerning our manuscript entitled “Is it feasible to preserve a self-sustaining population of Yangtze finless porpoise in the highest density section of Yangtze River?”(ID: water-2103222). Those comments are all valuable and very helpful for revising and improving our paper, as well as the important guiding significance to our researches. We have studied comments carefully and have made correction which we hope meet with approval. Revised portion are marked in red in the paper. The main corrections in the paper and the responds to the reviewer’s comments are as flowing:

Responds to the reviewer’s comments: 

  1. Response to comment: (also if this parameter “sex ratio at birth” cannot be affected by conservation measures)

Response: It is really true as Reviewer suggested that it is generally accepted that the sex ratio at birth cannot be affected by conservation measures. But there have been rumors about the Yangtze finless porpoise, the newborn sex ratio of Yangtze finless porpoise has shifted during ex-situ conservation; According to Tian's 2011 Master's thesis, precipitation was negatively correlated with the sex ratio at birth in wild Taihang Mountains macaques (R = -0.769, p = 0.074). Therefore, we hypothesized that protective measures might affect the sex ratio at birth of the finless porpoise.

  1. Response to comment:(of what vital stages?)

Response: We are very sorry for our negligence. It is juvenile Yangtze finless porpoises (age 0 to 1 and age 1 to 2)

  1. Response to comment:(generally, a population having 99% probability to survive over 1000y can be considered to have the minimum viable population size, therefore this analysis could be extended over 1000y) Response: Considering the Reviewer’s suggestion, we have simulated the trends over 1000 years. The rapid decline of the Yangtze finless porpoise population between 2006 and 2012 has been largely curbed, but the species is still critically endangered and conservation efforts need to be strengthened. As a result, millennium-scale assessments may currently have limited practical implications for the conservation of the Yangtze finless porpoise, as shown in the figure below.

  1. Response to comment:(was established for females at 4 and for males at 5 years)

Response: We are very sorry for our negligence. In this study, the average age of first-time offspring was 4 years for females and 5 years for males. 5. Response to comment:(Li, 2017, therefore, for this modeling)

Response: For this model, the sex ratio at birth was 1:1 in the baseline scenario.

  1. Response to comment:(Thus)

Response: We are very sorry for our incorrect writing. This has been amended “thus”.

  1. Response to comment:(while exchange of individulas of 2-3 years only? This sentence is not clear to me)

Response: We have re-written this part according to the Reviewer’s suggestion. Annually capture and supplement a certain amount of 2-3 aged individuals with a sex ratio of 1:1 was set to simulate individual exchange; After the 2nd age of the Yangtze finless porpoise, the relationship between mother-calf pairs is not very close. The exchange of the 2nd-3rd age porpoise can ensure the survival rate of the exchange individual, it can also minimize the effect of individual exchange on the reproductive capacity of the original population.

  1. Response to comment:(than genetic…?)

Response: We are very sorry for our negligence. This has been amended “The percentage of adult females breeding rate, sex ratio at birth and mortality from age 0 to 1 yielded the highest sensitivity of Stoch-r, whereas for final population size (N-all), the results were similar; But it is more sensitive to carrying capacity than mortality from age 0 to 1 (figure 2-3).”

  1. Response to comment:(all these scenatios were simulated over 100 years, I guess, please simulate the trends over 1000 years and add a graphic of the temporal trends over time to your manuscript)

Response: Considering the Reviewer’s suggestion, we have simulated the trends over 1000 years. The rapid decline of the Yangtze finless porpoise population between 2006 and 2012 has been largely curbed, but the species is still critically endangered and conservation efforts need to be strengthened. As a result, millennium-scale assessments may currently have limited practical implications for the conservation of the Yangtze finless porpoise, as shown in the figure below.

  1. Response to comment:(was greater)

Response: We have re-written this part according to the Reviewer’s suggestion. This has been amended “when the immigration reached 10, Stoch-r was greater than 0, the population size showed an increasing trend”

  1. Response to comment:( goes … gone ranges …ranged is better) Response: We have made correction according to the Reviewer’s comments. This has been amended “When the number of individuals exchanged goes from 0 to 8, Stoch-r gone ranges -0.0385 to -0.0367; when the number of individuals exchanged goes from 8 to 16, Stoch-r ranged -0.0367 to -0.0244; When the number of individuals exchanged goes from 0 to 16, Stoch-r gone ranges -0.0385 to -0.0244; When the number of individuals exchanged goes from 16 to 32, Stoch-r ranged -0.0244 to 0.0006, results were presented in Table 2.”
  2. Response to comment:( explains)

Response: We are very sorry for our incorrect writing. This has been amended “explains”.

  1. Response to comment:( construction is not clear to me, please explain better what do you want to mean…)

Response: We are very sorry for our incorrect writing. This has been amended “Construction of wading projects”.

  1. Response to comment:( reproductive output? Reproduction rate?) Response: We have re-written this part according to the Reviewer’s suggestion. This has been amended “Reproduction rate is the determining factor of how many descendants will exist in a population”.
  2. Response to comment:( May these two papers: Demography of the bottlenose dolphin Tursiops? truncatus (Mammalia: Delphinidae) in the Eastern Ligurian Sea (NW Mediterranean): quantification of female reproductive parameters, The European Zoological Journal; Reproductive biology of female common dolphins (Delphinus delphis) in New Zealand waters. Mar Biol)

Response: We have made correction according to the Reviewer’s comments. The variability in ovulation rate as determined by corpora count may suggest that some females ovulate more often than others (Palmer et al. 2022). Therefore, it is an important task to excavate and protect the fine breeding porpoise germplasm. Long-term studies have greatly enhanced the possibility for collecting relevant demographic information on long-lived, slowly reproducing cetacean populations in the wild (Rossi et al. 2017).

  1. Response to comment:( main)

Response: We are very sorry for our incorrect writing. This has been amended “main”.

  1. Response to comment:( Please, add to this legend the explanation of the acronyms used)

Response: Considering the Reviewer’s suggestion, this has been added “AFB: adult females breeding; MA0: mortality from age 0 to 1; MA1: mortality from age 1 to 2; SRB: sex ratio at birth; IPS: initial population size; K: carrying capacity; MBP: males in breeding pool; CS: catastrophes severity.”

Special thanks to you for your good comments.

Reviewer 2 Report

Several studies on Population Viability Analysis (PVA) are available on Yangtze finless porpoise and the present study is just an addition to that. While there is no issue regarding reassessment of the PVA of the threatened taxa, not giving due credit to the earlier studies and explaining how the current study makes further contribution for conservation ecology of the species does not reflect well on authors.

There are several studies on PVA of Yangtze finless porpoise. While some studies are cited by the authors, other studies are not. Surprisingly, authors have not cited their own study (Bin et al. 2021 Indian J Animal Res 55:1515-1520), which talks about the same issues. In fact, authors make the same conclusion “... with the implementation of policies such as a total ban on productive fishing in the main stream of the Yangtze River, they could enhance the protection of YFP population”, which is in the title of Bin et al. (2021). Although reassessment if PVA for this threatened organism is not an issue, not being completely honest and not citing earlier studies appropriately is a scientific misconduct.

Authors should add a separate paragraph in the introduction to highlight all the earlier studies on PVA of the species and explain why there is a need to reassess the PVA of the species and how the current study contributes additional analysis for conservation ecology of the species. Authors should cite their earlier work on the species and explain the further progress made in the current study. Reiterating similar conclusion in multiple studies on the same species and area constitutes the scientific misconduct called as self-plagiarism.

Graphs of population size changes for the baseline model and other models used for the study will be helpful.

Author Response

Thank you for your letter and for the reviewers’ comments concerning our manuscript entitled “Is it feasible to preserve a self-sustaining population of Yangtze finless porpoise in the highest density section of Yangtze River?”(ID: water-2103222). Those comments are all valuable and very helpful for revising and improving our paper, as well as the important guiding significance to our researches. The main objective of this article is actually to explore the creation of a self-sustaining population, based on the local exchange of ex-situ individuals. Other articles on the viability of the Yangtze finless porpoise population were cited, but not this article on the 2021 you mentioned, in order to avoid the impression that it was entirely self-referential; The second is that the article is actually a whole study, and the guidance for the protection of small populations is not strong, and this article is mainly to explore in-situ and ex-situ integration applications to solve the small population protection problem. If you feel it is necessary, we would be happy to quote it.

We have studied comments carefully and have made correction which we hope meet with approval. Revised portion are marked in red in the paper. The main corrections in the paper and the responds to the reviewer’s comments are as flowing:

Responds to the reviewer’s comments: 

  1. Response to comment: (also if this parameter “sex ratio at birth” cannot be affected by conservation measures)

Response: It is really true as Reviewer suggested that it is generally accepted that the sex ratio at birth cannot be affected by conservation measures. But there have been rumors about the Yangtze finless porpoise, the newborn sex ratio of Yangtze finless porpoise has shifted during ex-situ conservation; According to Tian's 2011 Master's thesis, precipitation was negatively correlated with the sex ratio at birth in wild Taihang Mountains macaques (R = -0.769, p = 0.074). Therefore, we hypothesized that protective measures might affect the sex ratio at birth of the finless porpoise.

  1. Response to comment:(of what vital stages?)

Response: We are very sorry for our negligence. It is juvenile Yangtze finless porpoises (age 0 to 1 and age 1 to 2)

  1. Response to comment:(generally, a population having 99% probability to survive over 1000y can be considered to have the minimum viable population size, therefore this analysis could be extended over 1000y) Response: Considering the Reviewer’s suggestion, we have simulated the trends over 1000 years. The rapid decline of the Yangtze finless porpoise population between 2006 and 2012 has been largely curbed, but the species is still critically endangered and conservation efforts need to be strengthened. As a result, millennium-scale assessments may currently have limited practical implications for the conservation of the Yangtze finless porpoise, as shown in the figure below.

  1. Response to comment:(was established for females at 4 and for males at 5 years)

Response: We are very sorry for our negligence. In this study, the average age of first-time offspring was 4 years for females and 5 years for males. 5. Response to comment:(Li, 2017, therefore, for this modeling)

Response: For this model, the sex ratio at birth was 1:1 in the baseline scenario.

  1. Response to comment:(Thus)

Response: We are very sorry for our incorrect writing. This has been amended “thus”.

  1. Response to comment:(while exchange of individulas of 2-3 years only? This sentence is not clear to me)

Response: We have re-written this part according to the Reviewer’s suggestion. Annually capture and supplement a certain amount of 2-3 aged individuals with a sex ratio of 1:1 was set to simulate individual exchange; After the 2nd age of the Yangtze finless porpoise, the relationship between mother-calf pairs is not very close. The exchange of the 2nd-3rd age porpoise can ensure the survival rate of the exchange individual, it can also minimize the effect of individual exchange on the reproductive capacity of the original population.

  1. Response to comment:(than genetic…?)

Response: We are very sorry for our negligence. This has been amended “The percentage of adult females breeding rate, sex ratio at birth and mortality from age 0 to 1 yielded the highest sensitivity of Stoch-r, whereas for final population size (N-all), the results were similar; But it is more sensitive to carrying capacity than mortality from age 0 to 1 (figure 2-3).”

  1. Response to comment:(all these scenatios were simulated over 100 years, I guess, please simulate the trends over 1000 years and add a graphic of the temporal trends over time to your manuscript)

Response: Considering the Reviewer’s suggestion, we have simulated the trends over 1000 years. The rapid decline of the Yangtze finless porpoise population between 2006 and 2012 has been largely curbed, but the species is still critically endangered and conservation efforts need to be strengthened. As a result, millennium-scale assessments may currently have limited practical implications for the conservation of the Yangtze finless porpoise, as shown in the figure below.

  1. Response to comment:(was greater)

Response: We have re-written this part according to the Reviewer’s suggestion. This has been amended “when the immigration reached 10, Stoch-r was greater than 0, the population size showed an increasing trend”

  1. Response to comment:( goes … gone ranges …ranged is better) Response: We have made correction according to the Reviewer’s comments. This has been amended “When the number of individuals exchanged goes from 0 to 8, Stoch-r gone ranges -0.0385 to -0.0367; when the number of individuals exchanged goes from 8 to 16, Stoch-r ranged -0.0367 to -0.0244; When the number of individuals exchanged goes from 0 to 16, Stoch-r gone ranges -0.0385 to -0.0244; When the number of individuals exchanged goes from 16 to 32, Stoch-r ranged -0.0244 to 0.0006, results were presented in Table 2.”
  2. Response to comment:( explains)

Response: We are very sorry for our incorrect writing. This has been amended “explains”.

  1. Response to comment:( construction is not clear to me, please explain better what do you want to mean…)

Response: We are very sorry for our incorrect writing. This has been amended “Construction of wading projects”.

  1. Response to comment:( reproductive output? Reproduction rate?) Response: We have re-written this part according to the Reviewer’s suggestion. This has been amended “Reproduction rate is the determining factor of how many descendants will exist in a population”.
  2. Response to comment:( May these two papers: Demography of the bottlenose dolphin Tursiops? truncatus (Mammalia: Delphinidae) in the Eastern Ligurian Sea (NW Mediterranean): quantification of female reproductive parameters, The European Zoological Journal; Reproductive biology of female common dolphins (Delphinus delphis) in New Zealand waters. Mar Biol)

Response: We have made correction according to the Reviewer’s comments. The variability in ovulation rate as determined by corpora count may suggest that some females ovulate more often than others (Palmer et al. 2022). Therefore, it is an important task to excavate and protect the fine breeding porpoise germplasm. Long-term studies have greatly enhanced the possibility for collecting relevant demographic information on long-lived, slowly reproducing cetacean populations in the wild (Rossi et al. 2017).

  1. Response to comment:( main)

Response: We are very sorry for our incorrect writing. This has been amended “main”.

  1. Response to comment:( Please, add to this legend the explanation of the acronyms used)

Response: Considering the Reviewer’s suggestion, this has been added “AFB: adult females breeding; MA0: mortality from age 0 to 1; MA1: mortality from age 1 to 2; SRB: sex ratio at birth; IPS: initial population size; K: carrying capacity; MBP: males in breeding pool; CS: catastrophes severity.”

Special thanks to you for your good comments.

Reviewer 3 Report

We greatly appreciated the authors' effort to provide a sound scientific tool for the protection and conservation management of the threatened Yangtze finless porpoise population. However, the work fails to provide a clear methodological picture of the analysis performed. The results and discussions lack fluency and clarity and are sometimes confusing. However, I feel that the work deserves further attention, so I look forward to an improved version of the manuscript. Please find attached the comments I made in the ms.

Author Response

(The authors gave the same response as above.)

Round 2

Reviewer 2 Report

Authors have not done the revision satisfactorily. I specifically suggested that they should cite their paper published in 2021 (Bin et al. 2021 Indian J Animal Res 55:1515-1520) as there are ethical issues. Authors have cleverly refused to cite their paper by saying that this will lead to self-citation. However, they are definitely not too shy of self-citation, as can be seen from their reference list. Complete transparency is important aspect of scientific communication. If authors do not want to be criticized of duplicate publications, they should cite their 2021 paper and explain how the current study differ from the 2021 paper. As suggested in the previous comments, authors should add a separate paragraph in the introduction section where they cite all the earlier PVAs on Neophocaena asiaeorientalis and explain the rationale for performing the current study. This is beneficial for the authors, so as to avoid post publication criticism of multiple similar articles. In the discussion section, authors should again refer to their study and explain how the current study differ from or add to the outputs of the previous study.

Author Response

Dear Editors and Reviewers:

Thank you for your letter and for the reviewers’ comments concerning our manuscript entitled “Is it feasible to preserve a self-sustaining population of Yangtze finless porpoise in the highest density section of Yangtze River?”(ID: water-2103222). Those comments are all valuable and very helpful for revising and improving our paper, as well as the important guiding significance to our researches. We have studied comments carefully and have made correction which we hope meet with approval. Revised portion are marked in red in the paper. The main corrections in the paper and the responds to the reviewer’s comments are as flowing:

Responds to the reviewer’s comments:

1.Response to comment: (authors should add a separate paragraph in the introduction section where they cite all the earlier PVAs on Neophocaena asiaeorientalis and explain the rationale for performing the current study)

Response: We have re-written this part according to the Reviewer’s suggestion. “Based on the population viability analysis, researchers pointed out that under the comprehensive protection measures such as the prohibition of productive fishing, the population of YFP in the main stream of the Yangtze River will increase (Wu et al. 2021). However, the YFP in the main stream of the Yangtze River belongs to frag-mented distribution, so there is a large error in the analysis of the whole as a population, and there is only the significance of macro judgment of the population trend; The depth of understanding for each small population is inconsistent, and it is also difficult to accurately estimate the relevant analysis parameters. Therefore, selecting relatively independent and well-understood small populations for correlation analysis has strong guidance for the protection of regional small populations. At the same time, the model analysis results and the estimation of relevant parameters can also be verified and corrected in the continuous protection practice.”

2.Response to comment: In the discussion section, authors should again refer to their study and explain how the current study differ from or add to the outputs of the previous study.”

Response: We are very sorry for our negligence. This has been amended “Thus, with the implementation of policies such as a total ban on productive fishing in the main stream of the Yangtze River, they could enhance the protection of YFP population (Wu et al., 2021).”

Reviewer 3 Report

The paper still needs improvements.

First, please proofread the English in order to make the paper readable.

Please, show the stochastic parameters tested under the stochastic model. The differences between the parameters defining the deterministic and stochastic models are not clear! Please, try to put these in evidence. please see supplementary comments in the text attached. 

Author Response

Dear Editors and Reviewers:

Thank you for your letter and for the reviewers’ comments concerning our manuscript entitled “Is it feasible to preserve a self-sustaining population of Yangtze finless porpoise in the highest density section of Yangtze River?”(ID: water-2103222). Those comments are all valuable and very helpful for revising and improving our paper, as well as the important guiding significance to our researches. We have studied comments carefully and have made correction which we hope meet with approval. Revised portion are marked in red in the paper. The main corrections in the paper and the responds to the reviewer’s comments are as flowing:

Responds to the reviewer’s comments:

1.Response to comment: (please explain in few words what means independent in the context)

Response: We have re-written this part according to the Reviewer’s suggestion.“which is now considered as an independent species, population genomics of YFP reveal an incipient cetacean species adapted to freshwater”

  1. Response to comment:(too vague)

Response: We are very sorry for our negligence. This has been amended “The populations of YFP in Yangtze River mainstream, Dongting Lake and Poyang Lake have stable size and distribution. ”

  1. Response to comment:( what's the relevance? all porpoises have such a system!)

Response: We are very sorry for our negligence. The decrease of quiet water environment is one of the important factors affecting finless porpoise. This has been amended“The YFP has developed sonar system, which carries out life activities such as cruising, predation and communication through high frequency pulse signal and low frequency continuous signal. With the rapid social and economic development in the Yangtze River Basin, the quiet water environment is becoming less and less, the living space of the YFP is increasingly compressed.”

  1. Response to comment:(??)

Response: We are very sorry for our incorrect writing. This has been deleted “The flow pattern is the core content of the habitat hydrological environment ”.

  1. Response to comment:( what do you mean? do you refer to the simulation of translocations? Please rephrase!)

Response: We are very sorry for our incorrect writing. This has been amended “Periodic translocations of protected objects is an important intensive management approach to supplement the single viable population”.

  1. Response to comment:( Please rephrase!)

Response: We have re-written this part according to the Reviewer’s suggestion. “However, VORTEX has relatively few requirements for data, and the estimation results are relatively reliable. The model is widely used (Lacy. 2000).”

  1. Response to comment:( what do you mean? median predicted the time?)

Response: We are very sorry for our negligence. This has been amended “Recent studies have shown that the median time to extinction of YFP was 25–33 years in the Yangtze River”

  1. Response to comment:( is this a high value? please explain!)

Response: Public data show that in 1990, three females, two males and five finless porpoises were put into the Tian'ezhou ex-situ conservation area for trial breeding, and in 2021, the census has developed to 101 finless porpoises. Therefore, we believe that the inbreeding recession of the Yangtze finless porpoise may not be so serious, and the predecessor (Zhang et al, 1999) also took 3.14.

  1. Response to comment:( what do you mean by paternity lest? what is the relevance?)

Response: The parent-child relationship test can provide us with the breeding information of the finless porpoise, for example, the age difference between mother and child reflects the upper limit of its minimum reproductive age

  1. Response to comment:(what variables explain the differences?) Response: Different simulation scenarios are described in the material methods and tables.
  2. Response to comment:( ??)

Response: We are very sorry for our incorrect writing. This has been amended “providing a good nursing environment for mother-calf pairs”.

  1. Response to comment:( to which model do you refer precisely?)

Response: We are very sorry for our incorrect writing. This has been amended “These reduction in gene flow resulted in a decrease in the stochastic growth rate (See Model 8-13 in Table 2 for details).”

  1. Response to comment:( ???)

Response: We are very sorry for our incorrect writing. This has been amended “In the conservation scenarios,”.

  1. Response to comment:( Please rephrase!)

 Response: We are very sorry for our incorrect writing. This has been amended “It is suggested to improve the living environment "small but comprehensive" by ana-lyzing the sensitivity parameters, and formulate a complete set of protection schemes may be the main inspiration to us from the population viability analysis of YFP popu-lation in HMS.”

Special thanks to you for your good comments.
